# Fire use practices, knowledge and perceptions in a West African savanna parkland

Esther Ekua Amoako[1,2]*, James Gambiza[1]

**1** Department of Environmental Science, Rhodes University, Grahamstown, South Africa, **2** Department of Ecotourism and Environmental Management, University for Development Studies, Tamale, Ghana

* eamoako@uds.edu.gh

**Data Availability Statement:** All relevant data are contained within the manuscript and a Zipfile of fire counts uploaded.

**Funding:** The authors received no specific funding this research.

## Abstract

Understanding people's practices, knowledge and perceptions of the use of fire and fire regimes can inform fire management plans that could contribute to savanna conservation and sustainable management. We investigated the frequency of fire use, control and perceptions of fire regimes for selected livelihood and socio-cultural activities in six districts in the Guinea savanna of Ghana. The six districts were selected to have a good representation of fire prone areas in the region based on fire frequency data obtained from the CSIR Meraka Institute, South Africa. A multiple regression analysis showed that people's use of fire for the selected socio-cultural activities from district, occupation, gender, age and ethnic group significantly predicted fire use for the activities $R^2 = 0.043$, $F_{(5,498)} = 5.43$, $p < 0.000$. Age and occupation added significantly to the use of fire. The study revealed that the majority of respondents (83%) across the study districts used fire once a year for at least one of the following activities: land preparation, weed/pest control, burning postharvest stubble, bush clearing around homesteads, firebreaks, charcoal burning and hunting. The study also showed a higher frequency of fire use for land preparation for cropping than for the other activities. Less than a fifth of the respondents (17%) indicated that they do not use fire for any of the selected activities. The majority of respondents (65%) mentioned that they controlled their use of fire to prevent destruction to property or injuring humans. The study revealed a higher frequency of fire use in the dry season for land preparation for cropping. However, respondents rated season of burning as the most important attribute, with little attention to the other attributes of a fire regime, contrary to what is theoretically recognized. Understanding traditional fire use practices in terms of how to regulate the mix of frequency, intensity/severity, season, size and type of fire for these and other socio-cultural purposes could help to mitigate and/or manage bushfires in West African savannas and enhance savanna conservation and management. Hence, the need to better understand people's knowledge and perceptions of fire regimes in fire assisted socio-cultural practices in West Africa.

**Competing interests:** The authors have declared that no competing interest exist.

## Introduction

Historically, humans in many regions have used fire extensively for various land use practices such as agriculture, hunting, foraging and pasture management, which has created diverse habitats and landscapes [1–5]. The use of fire may be prescribed, controlled, uncontrolled or unintentional, depending on the source of the fire and the purpose of use [6,7]. It is reported that up to 7% of the world's population use slash and burn agriculture which has been practised for about 12,000 years [8,9]. Adar-burning, fire fallow, chitemene and swidden are all terms used for the slash and burn processes used during land preparation before cropping in different regions of the world [6,9,10]. The practical details of slash and burn may differ from one place to another depending on local social and environmental conditions, but the purpose is mainly for crop and animal husbandry.

Traditional agriculture and other land-based socio-economic activities (e.g., honey hunting, game hunting and charcoal making) have been the source of food supply and income for both rural and urban economies of most countries of sub-Saharan Africa [2,11] In fire-prone West African savannas, use of fire is integral to socio-cultural practices that are important for rural livelihoods and associated socio-economic activities [12,13]. The savannas provide land resources referred to as agroforestry parklands where the bulk of cereals, grains and livestock are produced, usually by means of traditional methods of agriculture [14–16].

The parkland system is a traditional system of agro-silvo-pastoral system where crops are cultivated and animals kept among indigenous tree species. Fire is used in these parklands to clear the land of vegetation for cropping, to burn postharvest stubble, as well as for weed management [17–19]. Also, pastoralists in these parklands and woodland savannas have often used fire to control pests and to stimulate fresh grass growth in the early part of the dry season [2,20]. Many rural people in these savannas still depend on hunting as a traditional livelihood strategies to supplement household income and protein needs [19,21,22]. A common method of hunting in the savanna is the use of fire to flush out animals or by burning early in the dry season to attract animals to the new growth where they can be hunted [23]. Other studies have reported that fire is also caused by honey hunters, arson, careless disposal of cigarettes and children trying to mimic their parents use of fire [6].

Similar to traditional burning in Australia and some parts of the world, [24] explains that, these fires are used to achieve multiple livelihood purposes. The fires are set to whole farmlands for cropping, woodlands to open up new areas for new farms, as well as for hunting. The fires consume dead and dying grasses, tree litter, shrubs, small trees. In West Africa, these fires are usually difficult to control during the Harmattan (North-Easterly wind characterised by very low humidity and relatively high windy conditions occurring between November and April). Fires left uncontrolled during this period, may result in large and persistent fires [25,26].

Satellite images of the West Africa savanna, as well as ground truthing, have shown that burning reaches its peak in the early to mid-dry season (November to January) [27,28]. [27] asserts that late season fires are observed in agricultural lands and wetlands. Thus, the evidence is that the fires are anthropogenic [26,29,30] and are usually made on small farms for cropping, as well as for hunting, charcoal making and other socio-cultural activities. Small fires from croplands in Africa, for instance, accounted for 41% of overall global burnt area per annum [30]. Thus, these fires are likely to have significant effects of fire impact at the landscape level and on the global environment [27].

There are some regulations on bushfires in West Africa, that range from restrictive to less restrictive. The P.N.D.C. Law 229 (Bush Fire Prevention and Control Law, 1990) of Ghana for instance, indicates that if an action of a person results in the uncontrolled burning of a farm, forest or grassland, the person is liable on conviction to a fine [31]. The fine is 1. Not less than

two hundred and fifty penalty units and not more than one thousand penalty units or 2. To a term of imprisonment or community labour not exceeding twelve months or 3. To both the fine and the imprisonment or community labour. For a subsequent offence it can be a term of imprisonment or community labour not exceeding two years. However, the view among environmentalists in Ghana, is that the implementation of the law has not been proactive. Thus, indiscriminate bush burning continues to be practiced in most rural areas of Ghana [31,32]. The challenges in the implementation of the law have been attributed to the lack of resources by the Forestry Commission and the Environmental Protection Agency to enforce the legislation, as well as the lack of stakeholder involvement in the enactment of the law [31]. However, a few district assemblies in collaboration with the Agricultural Extension and Education unit of the Ministry of Agriculture, non-governmental organisations through non-formal education services have chalked some successes in implementing no-burning by-laws through the formation fire management squads in some communities in the Upper East and Northern regions of Ghana [31].

In neighbouring Burkina Faso, there are decrees on the use of fire in rural lands which make precise authorisation or prohibition of the use of fire according to the circumstances. Bushfires are prohibited, but management fires (controlled burning for agricultural purposes) and customary fires managed by a traditional authority and village land management committees are allowed [33]. These decrees have worked quite well since 1997, because the traditional authority has played key roles in the management of fires in the country [33].

Studies on fire use in savannas have focused on the management and impact of fire on the environment [34–38]. However, some research has focused on understanding people's knowledge, practices and perceptions of 'fire-assisted' livelihood activities [19,24,39–43]. Nevertheless, the extensive knowledge, values, practices and perceptions of the traditional uses of fire in subsistence agriculture, as practiced in tropical savannas in Africa, and West African agroforestry parklands in particular, have been under estimated and thus received limited research attention.

Thus, in consonance with Eriksen's (2007) research on "Why do they burn the bush"? It was important to gather information on people's knowledge and perceptions on fire in the Guinea savanna of West Africa where traditional agriculture is embedded in the parkland system. Understanding how people perceive the role of fire in their socio-cultural and livelihood activities is necessary to inform fire management plans so that they are well-matched with local knowledge, fire use practices, as well as traditional land use systems [44–47]. This could contribute to conservation and sustainable management of savannas of Africa. Thus, the objective of the study was to investigate peoples' practices, knowledge and perceptions of the traditional uses of fire. Answers were sought to:

1. What are the relationships between fire use and the selected demographic and socio-cultural activities?

2. How frequently is fire used for the selected socio-cultural activity?

3. How often is fire controlled for the selected activities?

   What are people's perceptions and knowledge of a fire regime as defined by [48] and [49]?

## Materials and methods

### Background and setting of the study area

The study was conducted in the Northern region of Ghana, located in the Guinea savanna ecological zone (9.5439° N, 0.9057° W). The climate of the region is tropical with a unimodal rainfall distribution, with an annual mean rainfall of 1,100 mm [50]. Thus, the region has only one

cropping season that falls between May and October [50]. The peak of the rainy season ranges between July and September, with the rainfall exceeding potential evaporation over a relatively short period [51]. The mean annual temperature is 27˚C. The region experiences comparatively high annual potential open-water evaporation of 2,000 mm. Due to its proximity to the Sahel, the region experiences dry, dusty north-easterly winds (Harmattan) between November and April, which facilitates the annual vegetation burning [52] across all the districts in the region. The area is characterised by large areas of natural pastures with grass species from the sub-families of Andropogoneae and Paniceae [53,54], interspersed with fire and drought-resistant woody species from the families Fabaceae and Combretaceae [35,55].

**Population.** The region has a population of nearly 2.5 million people in 70,383 km$^2$, representing ten percent of the total population of the country [56]. The population density is 37 persons per km$^2$ [56]. The majority of the population lives in rural areas. The region has five paramount chiefdoms (traditional areas), namely, Dagomba, Gonja, Mamprusi, Mo and Nanumba. Each traditional area represents a major traditional group in the region [57]. The population of these groups varies, with the largest group being the Dagomba which constituting 30% of the regional population. The Gonja and Mamprusi groups comprise over 7% of the population [56].

**Agricultural systems.** Agriculture contributes more than 90% of household income and employs more than 70% of the population in the region [56]. Approximately 80 to 90% of all land in Ghana is customary land, giving the highest-ranked chief the absolute right of possession, transfer and lease. Agricultural lands in communities have been transferred from generation to generation within a clan. This customary land tenure system is recognised by the Constitution of Ghana [58].

The farming system is traditional agroforestry, which is usually a combination of growing food crops and keeping animals for multiple purposes. Farm sizes of one to two hectares [59] are located around one to six kilometres from the compound house [60]. Families may cultivate one or more farm holdings depending on the availability of land in the community. Crops are cultivated for about five to ten years interspersed among economically valuable indigenous tree species, a system of agriculture which is commonly referred to as agroforestry parklands [16]. The crop fields within parklands are allowed to rest for a fallow period of about three to five years to replenish soil fertility after several years of continuous cultivation.

Among the major crops grown are maize (*Zea mays*), millet (*Panicum miliaceum*), rice (*Oryza sativa*), yam (*Dioscorea* spp.), and various pulses and vegetables [61,62]. Cropping is also done around the home compounds in the rainy season and in valleys adjacent to water bodies, especially in the dry season [61]. These compound farms are usually permanent, because the soils are replenished by the continuous supply of household waste and livestock manure [60]. They are also burnt annually to control ticks and reptiles, as well as for visibility purposes [63]. There are considerable areas of parkland which have not been cultivated, either because of low soil fertility or they have been left fallow for a very long period [64].

The Northern region leads in livestock (ruminants and poultry) production in the country. Livestock is kept on both free range and semi-intensive systems by households and hired Fulani herdsman, respectively [60]. Other socio-economic activities are agro-processing (e.g., gari, rice and groundnut processing) and the processing of non-timber forest products (NTFPs) (e.g., shea butter processing).

## Data collection

Stratified, purposive and convenience sampling techniques were used. Data on daily fire counts (detected by sensors on Earth observation satellite) from 2013 to 2017 on Ghana, were

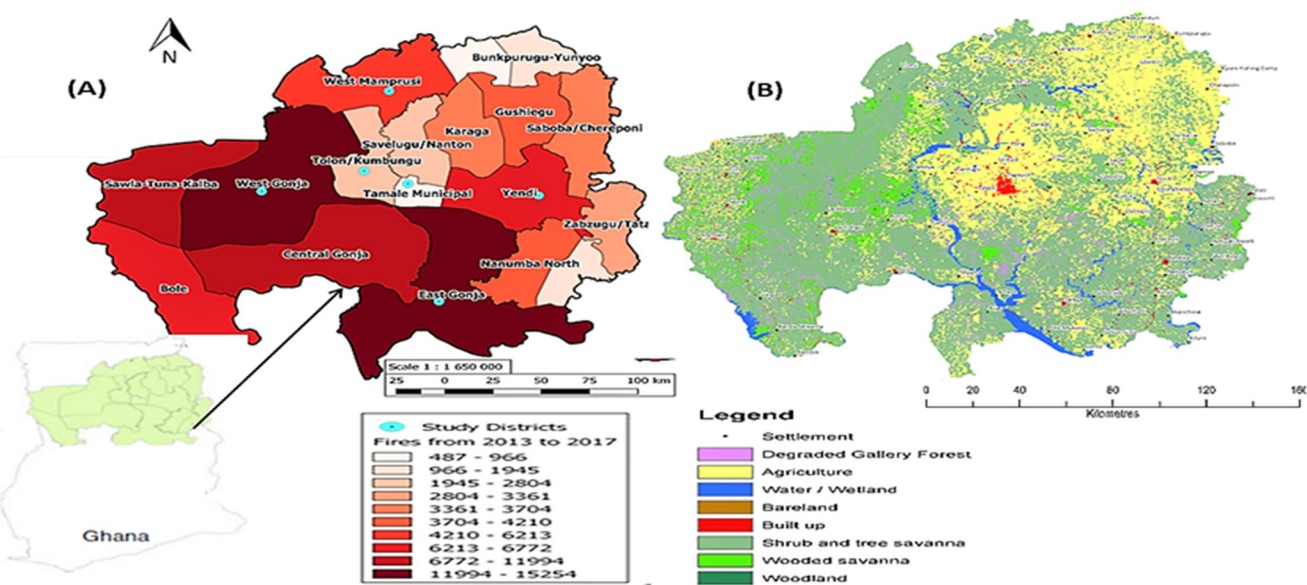

**Fig 1.** A-Fire frequency gradient (colour gradient), and selected study districts, B.- Land use map of study area for 2017. **A**–Map was created from fire count data (2013 to 2017) from ERORIC-AFIS under a CC BY license, with permission from EORIC, Ghana, original copyright (2017). **B**—Land use map —Republished from [CERSGIS-SERVIR] under a CC BY license, with permission from SERVIR, original copyright (2017).

received from EORIC (Earth Observation Research and Innovation Centre Ghana- (Earth Observation Research and Innovation Centre—https://eoric.uenr.edu.gh/?bunch_projects= fire-information-for-rural-farmers) in collaboration with the Advanced Fire Information the CSIR, Meraka Institute South Africa (http://www.csir.co.za). The first batch of fire data for the 2015 to 2017 was sent from EORIC, and data for 2013 and 2014 from the CSIR [65]. The fire count data for 2013 to 2017, was used to generate a fire map for the region (Fig 1A) which is compared to the land use/land cover map of the region (Fig 1B). The two maps show the link between fire frequency, vegetation and land uses in the Guinea savanna ecological zone.

To have a good representation of high, moderate and low fire prone areas in the region, six of the 18 districts and 10 study communities (Fig 2) were purposively selected for the survey with the help of the Regional Forestry office. The communities selected were Damongo Canteen and Mognori (West Gonja district), Kpalbe and Kushini (East Gonja District) were selected in the high fire frequency districts, Kata-Banawa (West Mamprusi district), and Kpligini (Yendi district) in the moderate fire frequency districts and Jagriguyilli and Nwodua in the Tolon, Kumbungu district, Tugu (Tamale South Municipality) in the low fire frequency districts. Tugu and Damongo Canteen are peri-urban communities, while the remaining eight are rural communities (Fig 2).

A survey was conducted by using a questionnaire drafted in English. People were asked questions in their respective languages on their perceptions, practices and knowledge of the use of fire on selected livelihood activities. The questionnaire had three main sections: 1. Demographic characteristics. 2. Traditional fire practices, knowledge and perceptions of the use of fire, and 3. Fire regime and fire control practices. The questions were mostly closed-ended. The closed-ended questions were three-point [66], and five-point Likert type questions, as well as binary type questions to indicate the importance and frequency of fire use [67]. Participants were given options to explain their choice of answers to some of the questions, to aid

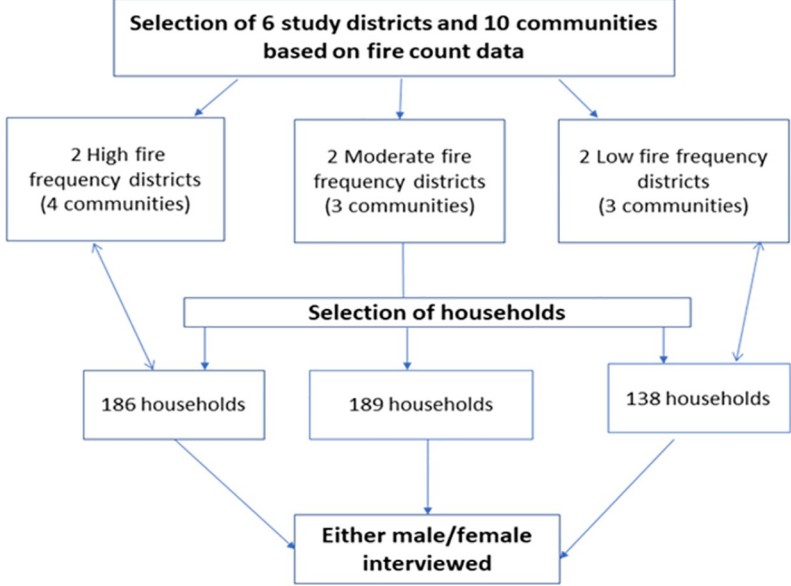

**Fig 2. Selection of communities and households for the survey.**

in guiding responses and facilitating data coding. The questionnaire yielded mainly nominal, ordinal and a few continuous variables (age and household size).

The open-ended questions were used to ascertain the activities for which people use fire, people's practices and knowledge of the use of fire, the impacts of fire on the environment, as well as their knowledge of fire management. The activities (land preparation, hunting, fire-breaks, bush clearing around homesteads, and burning stubble after harvest) were selected based on literature on fire use in savannas [19].

The data were collected with the assistance of a research team selected from the University for Development Studies in Tamale, who are fluent speakers of the language spoken in the selected districts. The research team were all males and comprised of five assistants for East and West Gonja Districts, four for West Mamprusi and five for Tolon-Kubungu, Yendi and Tamale South districts. They were trained on the objectives and some terminologies of the study.

We first met community leaders to seek permission and introduce the team by presenting kola nuts to the Chiefs or their representative at a brief meeting. We were permitted to engage with households with adults available because it was time for land preparation towards the farming season, thus some residents were not at home at most times of the interviews, so as we walked in the community, we select households with people available. However, either a male (mostly heads of households) or a female who was willing to answer the questions on fire use was interviewed in each of the communities.

[67] asserts that laws and government regulations are factors that may influence behaviours in responding to questions concerning themselves. Thus, we asked general questions on some sensitive items on hunting and charcoal making and sometimes probed further to get details. Data were collected from 11 March 2017 to 30 June 2017. The total number of responses was 532 of which 295 were from males and 237 were from females. About 20–30 minutes was used for a questionnaire depending on how fast the respondent understood and responded to the items on the questionnaire.

## Ethics statement

Formal Ethics Approval was not obtained. The research presents no risks of harm or no effects to the respondents, and involved no formal procedures for which written consents were required. However, verbal consents were sought from the community leadership by following community entry protocols via communal meetings and presenting traditional gifts (e.g., kola nuts) to the chiefs, the assembly persons and other leaders in the communities. Some of these communities the research team (particularly the lead author) have worked for several years in different domains of research other than anthropogenic fires. We informed the community members that participation was voluntary, and should anyone feel uncomfortable with any of the questions they could discontinue answering the questionnaires without any negative consequences. We also informed the respondents and the leaders of the communities that the data will be analysed anonymously and all the information will be stored in a secure space where only the lead researchers and the supervisor have access. Participants were made to understand that the responses were for an academic purpose and I showed my Identity Card to the assembly persons as a proof of my status as a student. Participants were also assured of confidentiality. All the interviews were conducted in the local languages of the selected districts as the majority of the respondents could not speak and understand English.

## Data analysis

The means of fire count data for the period under study were calculated and used to categorises the various hotspots in region. To provide an overview of socio-demographic characteristics, perceptions, knowledge and practices and respondents' use of fire, data were analysed using both descriptive and inferential statistics. Questions that yielded no and yes answers with reasons and some open-ended answers were classified into themes. All data were coded based on the commonalities in responses and data were entered, cleaned and analysed using SPSS Statistical Software Version 20.0. Some of the five-point Likert items were recoded into three-point items to allow more responses in the Likert items for statistical testing. Descriptive statistics were computed and a Chi-square test was used to examine for the presence of significant relationships and associations among variables across districts [67]. A Pearson's correlation between people's perceptions of fire effect on the environment and the frequency of fire use was also computed. In order to determine the relationship between fire use, socio-cultural practices and activities, multiple regression analyses were performed to examine how gender, occupation, district, age and ethnic group (which served as the independent variables) are correlated with fire use in the dry season, selected fire activities, reasons for fire use and fire control (as the dependent variables).

## Results

### Fire count data of study districts in the dry season

Eighteen districts in the Northern region of Ghana with data on fire counts were categorised into high fire frequency (6,213–15,254 counts), moderate fire frequency (2,804–6,213 counts) and low fire frequency (487–2,804 counts) districts. There were significant differences ($F_{5,80.5}$ = 7.4, $p < 0.001$) in mean dry season fire counts (i.e., from November to March) across the study districts over a period of five years (2013 to 2017). The highest mean dry season fire count for East Gonja (256.2±608.2) was 33 times higher than the counts in Tamale (7.83 ±19.22) which recorded the lowest fire counts (Table 1). Dry season fire counts in Tamale were significantly lower than counts in East Gonja ($p < 0.001$), West Gonja ($p < 0.01$), and Mion ($p = 0.01$) (Table 1).

**Table 1. Fire count from 2013–2017 (November to April).**

| District | Fire count per five (2013–2017) dry seasons (NOVEMBER-APRIL) | | |
|---|---|---|---|
| | Mean ±SD | Minimum | Maximum |
| Tamale | 7.8±19.22[a] | 38 | 122 |
| Tolon-Kumbungu | 52.9 ±12.7[b] | 343 | 585 |
| West Gonja | 226.7±686.8[c] | 2,689 | 3,758 |
| West Mamprusi | 81.5±75.6[ab] | 713 | 1,160 |
| Mion | 91.2 ±239[ab] | 619 | 786 |
| East Gonja | 256.2± 608.19[c] | 1,958 | 4,068 |

Mean values labelled with the same letter in a column were not significantly different at $p < 0.05$ Source: AFIS/CSIR Meraka Institute, South Africa. (Source: Amoako and Gambiza 2019).

Table 2 Fire count from 2013–2017 (November to April) Tamale were significantly lower than counts in East Gonja ($p < 0.001$), West Gonja ($p < 0.01$), and Mion ($p = 0.01$) (Table 1).

## Socio-demographic characteristics of respondents

The majority of respondents (79%) indicated they were farmers, 6% were students; and the remaining 15% engaged in other activities. Almost half of the of the respondents were female (45.5%) and 55.5% were male. The mean age of respondents was 39±15 years, with a minimum of 18 years and maximum age of 94 years. About a quarter of the respondents were between 26 and 32 years and the lowest (8%) were between of 47 and 53 years. The majority of respondents (66%) have no formal education and less than a quarter (15%) had education till Junior high school (JHS) or Middle school, and Senior High schools (SHS) (Table 2).

**Table 2. Gender, age and educational distribution of respondents.**

| Characteristic | Freq.(n) | % |
|---|---|---|
| **Gender** | | |
| Male | 295 | 55.5 |
| Female | 237 | 44.5 |
| Total | 532 | 100.0 |
| **Age (years) group** | | |
| 18–25 | 93 | 17.5 |
| 26–32 | 112 | 21.1 |
| 33–39 | 107 | 19.9 |
| 40–46 | 94 | 17.7 |
| 47–53 | 42 | 7.9 |
| 54+ | 84 | 15.8 |
| **Education** | | |
| No formal education | 347 | 65.5 |
| Primary | 46 | 8.7 |
| JHS/Middle school | 49 | 9.2 |
| SHS | 35 | 6.6 |
| Tertiary | 9 | 1.7 |
| Non-formal | 38 | 7.2 |
| Other | 6 | 1.1 |

The mean household (traditional household refers to a big house with several rooms belonging to the sons of (with the wives and children) one father or grandfather who is normally the head of the household. The house is normally built in a circular fashion with a big shared compound being the centre of the house) size was 12.2±8 persons, with a minimum of one and maximum of 70 persons. The respondents comprised of people from nine ethnic groups: Dagomba (43%), Mamprusi (33%), Gonja (16%), Fulani (2%), Frafra (1%), Komkomba (2%), Dagaaba (1%), Buno (1%) and Mossi (1%).

## Socio-demographic characteristics and fire use practices

The selected predictors in the regression analysis showed more significant relationships than other variables such as education and household size, thus, were excluded from further analyses. Significant, albeit weak relationships were found between fire use in the dry season and gender, age, ethnic group, occupation and district at $R^2 = 0.080$, $F_{(5,370)} = 6.35$, $p < 0.0001$. Three out of the five variables: district ($p < 0.001$), gender ($p = 0.007$) and occupation ($p < 0.05$) added significantly to fire use in the dry season. The highest contributing variable was district (0.214). Districts were categorised into low, moderate and high in relation to the fire count data that informed the selection of the districts, thus it is to be expected that this variable contributed most to the model. Gender (0.138) also contributed to the model with males having a higher likelihood of using fire in the dry season than females. Occupation (0.102) also contributed, with 79% respondents who indicated farming as their occupation. Age and ethnic group were not statistically significant (Table 3).

There was a weak significant relationship across demographic groups in terms of the different activities for which fire was used at $R^2 = 0.043$, $F_{(5,498)} = 5.43$, $p < 0.0001$. Of the five variables, age ($p < 0.0001$) and occupation ($p = 0.002$) added significantly to the use of fire for the selected activities in the region. Age as the highest predictor, contributed 0.180 while occupation contributed 0.144 to predict fire use for the selected activities. There were more people (79%) in farming than other occupations in this study. The remaining 15% in occupation including gari (a local food made from cassava) processing and only 6% were students. About 80% of the farmers were within the age brackets of 20 and 70 years with less than 20% below the age of 20. District contributed negatively (-0.071) to the model, while gender and ethnic group were not significant.

It was important to find out how the selected independent variables were correlated with people's reasons for fire use. The results were significant at $R^2 = 0.085$, $F_{(5,339)} = 6.23$, $p < 0.0001$. Three out of the five variables, namely ethnic group ($p < 0.001$), occupation ($p = 0.009$) and district ($p = 0.044$) added significantly to the model explaining reasons for fire use in the region. The highest contributing predictor was ethnic group (0.188). For instance, a total of 191 respondents indicated their use of fire for land preparation as part of their farming practices, with more Dagomba (64%) using fire for this activity than the Mamprusi (51%) and Mossi (1%). For occupation (0.149), again there were more Dagomba ethnic (40%) in farming than other ethnic groups such as the Fulani (4%) and district (0.110). Gender (0.030) and age (-0.043) were not significant.

Although, occupation ($p = 0.047$) contributed to the model for reasons why people control fire for the selected activities, the remaining four variables (age, gender, ethnic group and district) predicted poorly to the model and was therefore not statistically significant (Table 3).

## Frequency of fire use for selected activities

The majority of respondents (83%) indicated that they used fire for at least one of the selected activities: land preparation, weed/grass/pest control, burning stubble after harvest, bush

**Table 3. Multiple regression coefficients for the relationships between fire use practices and activities with some socio-demographic variables.**

| Variable | Unstandardized Coefficients | Standardized Coefficients | | t | Sig. | R2 | F-ratio | p | SEE | N |
|---|---|---|---|---|---|---|---|---|---|---|
| | B | Std. Error | Beta | | | | | | | |
| **Fire is use in the dry season** | | | | | | | | | | |
| (Constant) | -1.237 | 1.105 | | -1.120 | 0.263 | 0.088 | 6.35 | 0.000 | 3.75 | 370 |
| Gender | 1.075 | 0.396 | 0.138 | 2.712 | 0.007** | | | | | |
| Age range | 0.116 | 0.129 | 0.051 | 0.902 | 0.367 | | | | | |
| Occupation | 0.233 | 0.122 | 0.102 | 1.901 | 0.054* | | | | | |
| Ethnic group | 0.191 | 0.111 | 0.095 | 1.716 | 0.087 | | | | | |
| District | 0.602 | 0.147 | 0.214 | 4.087 | 0.000*** | | | | | |
| **Fire use for selected activities** | | | | | | | | | | |
| (Constant) | 0.94 | 0.093 | | 10.159 | 0.000 | 0.043 | 4.44 | 0.001 | 0.037 | 498 |
| Gender | -0.018 | 0.012 | -0.071 | -1.556 | 0.120 | | | | | |
| Age range | 0.040 | 0.034 | 0.053 | 1.191 | 0.234 | | | | | |
| Occupation | 0.040 | 0.011 | 0.180 | 3.759 | 0.000*** | | | | | |
| Ethnic Group | 0.032 | 0.011 | 0.144 | 3.040 | 0.002** | | | | | |
| District | 0.004 | 0.010 | 0.017 | 0.359 | 0.720 | | | | | |
| **Reasons for the use of fire** | | | | | | | | | | |
| (Constant) | 0.901 | 0.672 | | 1.342 | 0.181 | 0.085 | 6.25 | 0.000 | 2.22 | 339 |
| Gender | 0.178 | 0.088 | 0.110 | 2.026 | 0.044* | | | | | |
| Age range | 0.133 | 0.240 | 0.030 | 0.556 | 0.578 | | | | | |
| Occupation | -0.056 | 0.079 | -0.043 | -0.710 | 0.478 | | | | | |
| Ethnic group | 0.195 | 0.074 | 0.149 | 2.646 | 0.009** | | | | | |
| District | 0.216 | 0.066 | 0.188 | 3.252 | 0.001*** | | | | | |
| **Reasons for controlling fire** | | | | | | | | | | |
| (Constant) | 2.196 | 0.326 | | 6.735 | 0.000 | 0.025 | 1.8 | 0.108 | 1.09 | 361 |
| Gender | -0.019 | 0.116 | -0.009 | -0.163 | 0.871 | | | | | |
| Age range | -0.064 | 0.037 | -0.099 | -1.763 | 0.079 | | | | | |
| Occupation | -0.071 | 0.035 | -0.112 | -1.995 | 0.047* | | | | | |
| Ethnic group | 0.026 | 0.034 | 0.045 | 0.777 | 0.438 | | | | | |
| District | -0.055 | 0.041 | -0.073 | -1.337 | 0.182 | | | | | |

***$p < 0.001$,

**$p \leq 0.01$,

*$p \leq 0.05$.

clearing around homesteads, firebreaks, charcoal burning and hunting (Table 4). Less than a fifth (17%) of said that they do not use fire in these activities. There were varied responses on how often fire was used for the selected activities across the high, moderate and low fire frequency districts. There was a significant association ($\chi^2 = 39.5$, df = 9, $p < 0.001$) between fire frequency and land preparation for crop production amongst the three categorised districts. The majority of respondents in the high (86%), moderate (85%) and low (75%) fire frequency districts indicated they used fire once a year for land preparation, while only 8%, in both high and moderate and 2% in low fire frequency districts, used fire twice a year for land preparation (Table 3). On average, 14% of the respondents across the districts never used fire for land preparation.

There was a significant association between the frequency of fire used for weed control and the district ($\chi^2 = 66.6$, df = 9, $p < 0.001$). On average, 13% of the respondents used fire once a

**Table 4. Respondents' frequency of fire use for activities across the selected districts.**

| Activity | Frequency of fire use | District fire frequency | | | | χ2 | p-value |
|---|---|---|---|---|---|---|---|
| | | High n (%) | Moderate n (%) | Low n (%) | Pooled n (%) | | |
| Land preparation | Twice a year | 8 (8) | 8 (5) | 3 (2) | 19 (4) | 39.5 | 0.001 |
| | Once a year | 79 (86) | 180 (85) | 127 (75) | 386 (82) | | |
| | Never | 5 (6) | 22 (10) | 20 (24) | 67 (14) | | |
| Weed/grass /pest control | Twice a year | 4 (6) | 33 (12) | 1 (1) | 36 (10) | 66.6 | 0.001 |
| | Once a year | 8 (21) | 32 (11) | 8 (6) | 47 (13) | | |
| | Never | 28 (73) | 104 (77) | 140 (93) | 273 (77) | | |
| Burning stubbles after harvest | Twice a year | 4 (7) | 17 (9) | 1 (1) | 22 (6) | 74.8 | 0.001 |
| | Once a year | 24 (52) | 17(9) | 13 (9) | 54 (15) | | |
| | Never | 23 (41) | 129 (82) | 137 (91) | 289 (79) | | |
| Bush clearing around homesteads | Twice a year | 10 (9) | 23 (10) | 9 (11) | 49 (13) | 50.03 | 0.001 |
| | Once a year | 24 (46) | 35 (31) | 14 (9) | 73 (20) | | |
| | Never | 23(45) | 109 (59) | 122 (80) | 250 (67) | | |
| Hunting | Twice a year | 3 (8) | 20 (12) | - | 23 (6) | 26.9 | 0.003 |
| | Once a year | 5(7) | 8 (11) | 9 (6) | 32 (10) | | |
| | Never | 35(84) | 124 (77) | 139 (94) | 298 (84) | | |
| Firebreak | Twice a year | 1 (1) | 8 (3) | 3 (2) | 12 (3) | 159.7 | 0.001 |
| | Once a year | 55 (70) | 14 (27) | 43 (28) | 112 (29) | | |
| | Never | 22 (29) | 67 (70) | 110 (70) | 266 (68) | | |
| Charcoal making | Twice a year | 16 (30) | 24 (20) | 14 (9) | 54 (14) | 68.0 | 0.001 |
| | Once a year | 22 (38) | 8.5 (14) | 12 (16) | 63(16) | | |
| | Never | 19 (32) | 124 (66) | 116 (75) | 266 (70) | | |

year for weed control, while the majority (77%) never used fire for this purpose (Table 3). A little over a fifth (21%) of the respondents in the high-frequency zone, 11% in the moderate zone and 6% in the low zone, used fire once a year for weed control. In contrast, 73%, 77% and 93% of respondents in the high, moderate and low fire frequency zones, respectively, never used fire for weed control.

The majority of respondents (79%) never used fire to burn stubble after harvest, while less than a fifth (15%) used fire once a year for this purpose. More than half of the respondents in the high (52%) and low fire districts indicated that they burn stubbles after harvest. There was a significant association of frequency of fire use for stubble burning and districts ($\chi^2 = 74.8$, df = 9, p < 0.001).

Bush clearing around homesteads also showed a strong association ($\chi2 = 50.3$ df = 9, p < 0.001) with district. However, 67% of the respondents never used fire for clearing around homesteads. Twenty percent and 13% of the respondents used fire once and twice a year, respectively across the districts. However, about half (46%) of respondents in high fire frequency districts and 9% respondents in low fire frequency district said they used fire once a year. Whereas 45% in high, 60% in moderate and 80% in low fire frequency districts never used fire for clearing around homesteads (Table 3).

Respondents' use of fire to create firebreaks showed a strong significant association ($\chi^2 = 50.3$, df = 9, p < 0.001) among the districts. On average, 70% of the respondents in the moderate and low fire districts never used fire for firebreaks. About three-quarters (70%) of the respondents in high fire districts, and a little over a quarter of respondents in moderate (27%), as well as low (28%) fire frequency districts used fire once a year for firebreaks.

Similarly, there was a significant association of fire used for hunting ($\chi^2$ = 26.88, df = 9, p < 0.003) across the study districts. The majority (84%) of the respondents across the districts never used fire for hunting, with 11% in the moderate and as low as 7% in the high, 6% in the low fire frequency districts used fire once a year (Table 4).

## Fire control practice for selected activities

The majority of interviewees (72%) said they always controlled fire for land preparation. In contrast, the majority of respondents indicated they never controlled fire for the other selected activities (Table 4). Respondents who controlled fire and those who never controlled fire enumerated reasons for which fire used for the selected activities should be controlled. Thus, 62% of 532 mentioned that they control fire to prevent destruction to property, 29% said it was to prevent destruction to vegetation and wildlife, 6% mentioned to maintain soil fertility, and the remaining 3% indicated to prevent death or injury to humans.

There were significant associations between respondents' fire control practices and the selected activities across the three fire gradient districts ($\chi^2$ = 61.4, df = 9, p < 0.001). Around three-quarters (72%) of the respondents indicated that they controlled fire used for land preparation. Seventy-nine percent of the respondents in high fire frequency districts, and 74% for moderate and low (73%) indicated that they always controlled fire used for land preparation (Table 4). Less than a quarter across the district never controlled fire during land preparation.

Contrary to the control of fire for land preparation, 82% of the respondents in each of the districts never controlled fire used for weed control. Only a fifth of respondents in the high (10%) moderate (9%), and low (6%) fire frequency districts always controlled fire ($\chi^2$ = 48.5, df = 9, p < 0.001).

However, 59% of the respondents in the high fire frequency districts and less than a tenth of the respondents in the low (6%) and moderate (3%) fire districts, always controlled fire used for burning stubble. However, the majority (> 90%) of respondents in both moderate and low and the lowest 34% in high fire frequency districts never controlled fire used for burning stubble ($\chi^2$ = 39.5, df = 9, p < 0.001). As much as twice the number of respondents (65%) in high fire districts always controlled fire used for firebreaks than those in low fire frequency districts (31%), with only 6% in the moderate fire frequency districts ($\chi^2$ = 159.7, df = 9, p < 0.001).

The majority (90%) of respondents in all the districts indicated that they never controlled fire used for hunting, while less than a fifth of the respondents in the districts, controlled fire always (Table 5). However, respondents in the high fire (64%), low fire (23%) and moderate fire (16%) districts always controlled fire used for charcoal making (Table 5).

## Perceptions of the importance of fire regimes

There were varied perceptions of fire regimes. There was evidence of significant relationships in the perceptions of fire regimes with fire frequency districts (Table 6). Half of the respondents (50%) in the high, 43% in the low and 33% in the moderate fire frequency districts, were of the opinion that the season of burning was very important. While less than a fifth in each of the districts said the season of burning was not important.

The majority of respondents perceived the severity of fire as important but not very important: 45%, 41% and 28% in high, moderate and low districts, respectively, indicated that the severity of fire was important. However, more than half of the respondents (53%) in the low fire districts and less than half of the respondents in moderate (43%) and high (33%) fire frequency districts indicated that, the severity of fire was not important.

About 59% of the respondents perceived that the frequency of burn was unimportant. However, 33% in moderate, 30% in high and 17% low fire frequency districts showed the

**Table 5. Respondents' practice of fire control in socio-cultural fire use practices across the fire frequency districts.**

| Activity | Practice of fire control | District fire frequency | | | | χ² | p-value |
|---|---|---|---|---|---|---|---|
| | | High n (%) | Moderate n (%) | Low n (%) | Pooled n (%) | | |
| Land preparation | Always | 75 (79) | 125 (74) | 121 (73) | 321 (72) | 61.4 | 0.001 |
| | Sometimes | 12 (14) | 15 (12) | 10 (6) | 57 (13) | | |
| | Never | 6 (7) | 30 (15) | 34 (21) | 70 (16) | | |
| Weed//pest control | Always | 12 (15) | 19 (9) | 8 (6) | 26 (8) | 48.5 | 0.001 |
| | Sometimes | 8 (10) | 18 (8) | 1 (1) | 29 (9) | | |
| | Never | 7 (6) | 108 (83) | 132 (93) | 274 (83) | | |
| Burning stubbles after harvest | Always | 17 (59) | 4 (3) | 9 (7) | 30 (9) | 126.7 | 0.001 |
| | Sometimes | 6 (7) | 5 (3) | 4 (3) | 15 (5) | | |
| | Never | 28 (34) | 128 (95) | 129 (91) | 285 (86) | | |
| Bush clearing around homesteads | Always | 29 (65) | 30 (18) | 28 (20) | 87 (25) | 78.6 | 0.001 |
| | Sometimes | 6 (8) | 7 (35) | 1 (1) | 15 (5) | | |
| | Never | 22 (28) | 105 (79) | 114 (79) | 241 (70) | | |
| Hunting | Always | 9 (20) | 5 (2) | 6 (5) | 20 (6) | 67.4 | 0.001 |
| | Sometimes | 2 (5) | 10 (5) | 2 (1) | 14 (4) | | |
| | Never | 32 (75) | 131 (93) | 130 (94) | 293 (90) | | |
| Firebreak | Always | 42 (65) | 5 (6) | 45 (31) | 92 (27) | 171.9 | 0.001 |
| | Sometimes | 3 (4) | 1 (1) | 1 (1) | 6 (2) | | |
| | Never | 25 (31) | 115 (92) | 104 (68) | 244 (71) | | |
| Charcoal making | Always | 34 (64) | 21 (16) | 33 (23) | 88 (24) | 102.7 | 0.001 |
| | Sometimes | 4 (6) | 12 (5) | 1 (1) | 17 (5) | | |
| | Never | 24 (29) | 119 (79) | 110 (76) | 254 (71) | | |

frequency of burning was important. Similar to the responses on perceptions of fire frequency, on average the respondents (58%) in each of the districts were of the opinion that the type of fire was unimportant. Thus, less than a quarter of the number of respondents in each of the districts thought the type of fire was very important (Table 6).

**Table 6. Respondents' perceptions of the importance of fire regimes across the six fire districts.**

| Fire attribute | Importance of fire attributes | District fire frequency | | | | χ2 | p-value |
|---|---|---|---|---|---|---|---|
| | | High n (%) | Moderate n (%) | Low n (%) | Pooled n (%) | | |
| **Season of burning (timing; dry, season, early or late)** | Very Important | 46 (50) | 49(33) | 90 (53) | 185 (43) | 55.6 | **0.001** |
| | Important | 28 (32) | 95(51) | 49 (29) | 173 (40) | | |
| | Not important | 20 (18) | 23 (15) | 30 (18) | 73 (17) | | |
| **Severity of fire (what is burnt, soil vegetation mortality, high, moderate, low)** | Very Important | 21(21) | 28 (17) | 30 (18) | 79 (18) | 40.8 | **0.001** |
| | Important | 37(45) | 68 (41) | 47 (28) | 152 (36) | | |
| | Not important | 33 (34) | 75 (43) | 8 (53) | 196 (46) | | |
| **Frequency of fire (fire return interval and number of ignitions)** | Very Important | 14 (18) | 11 (5) | 24(15) | 49 (12) | 53.8 | **0.001** |
| | Important | 28 (30) | 62 (33) | 27(17) | 117 (29) | | |
| | Not important | 47 (52) | 79 (63) | 112 (69) | 238 (59) | | |
| **Type of fire (size and pattern)** | Very Important | 20 (22) | 16 (7) | 19 (12) | 55 (13) | 37.7 | **0.001** |
| | Important | 20 (22) | 62 (29) | 38 (23) | 120 (29) | | |
| | Not important | 48 (56) | 85(63) | 105 (65) | 240 (58) | | |

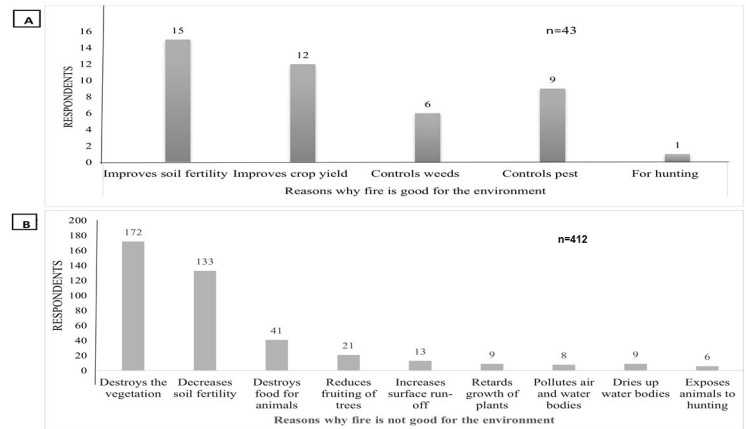

**Fig 3. A—Respondents' opinions on A—Positive effect of fire and B—Negative effects of fire on the environment.**

## Respondents' perceptions of the effects of fire on the environment and the frequency of fire use

About 77% of the respondents were of the view that fire was not good for the environment, 8% thought fire use was good for the environment while the remaining 14% indicated they did not know. However, 35% of the respondents who indicated that burning was good for the environment, perceived that burning increased soil fertility, followed by 28% thought that it improved crop yields, while less than 1% said it was good for hunting (Fig 3). In contrast, 42% of the respondents who said that burning was not good, were of the view that fire destroyed vegetation, and 33% felt that fire decreased soil fertility (Fig 3A and 3B).

## Discussion

### Socio-demographic characteristics and fire use practices

The five predictor variables with fire use in the dry season indicate that fire incidences around this time of the year is purposeful and that it is associated to reasons for fire occurrence in most rural savanna regions of Ghana. Thus, seasonality determines fire occurrence, which is also dependent on livelihood and some cultural activities in rural savannas of Africa [41,68]. This has also been found in fire prone areas around the world. For instance [69], listed activities carried out by native Americans during particular seasons of the year. The list of activities (e.g., clear agricultural fields and replenish the soil from the ash) in [69] aligns with [19] study in Kenya [70], in Mali as well as results in this study. Also, the Maori's of New Zealand primarily use fire for hunting game and clearing land for cropping [71] which were also identified in this study. Thus, the season for fire use is strongly related to the major occupation (farming in this case) of the people, ethnicity and location as found in the regression analysis.

There is an influence of location (districts) and ethnicity on fire activities in the dry season. This study stipulates that high fire frequency districts fall within areas with dense vegetation during the rainy season, which results in a build-up fuel during the dry season. Thus, people burn as they see drier grass cover during the dry season [68]. Districts with low fire frequency on the other hand, are characterised by sparse vegetation and more buildings than high fire frequency sites [65]. It is also noted that most of the respondents are indigenes of the various districts, hence corresponding to their occupation and other socio-cultural activities which are land-based, with a few who have migrated from other districts mainly for farming. The access to land

for farming through customary arrangements is very lenient in this part of Ghana [58]. Thus, land is transferred from one generation to the other, the same way as traditional practices and knowledge of fire use has been transferred over the years [6]. In rural Africa and Ghana for that matter, ethnicity can be linked to the location, as the Dagomba is the dominant ethnic group not only in this study but in the Northern region of Ghana [56,58]. The Dagomba ethnic group do not occupy the largest land area but were found in most of the communities particularly in the East and West Gonja (high fire frequency) districts but whose indigenes are the Gonja ethnic group. Ethnicity, however, did not make any significant prediction to the reason for the use of fire, indicating that all the ethnic groups have similar reasons for the use of fire, especially for land preparation for cropping which is the major reasons for fire use across all the districts.

As observed in the model, gender plays a role in fire activities. This is not surprising, because most of both men and women indicated farming as their major occupation. The women, however, specified that they were also engaged in other livelihood activities such as shea fruit picking and charcoal making which involve the use of fire. This is supported by anecdotal reports of fire use in parklands by women, to reduce the grass cover for easy picking of shea fruits and wood fuel (mainly a woman's role), as well as protection against harmful reptiles while engaged in these activities. Women may not be directly involved in the use of fire for some socio-economic and cultural practices such land preparation, hunting and clearing around homesteads due to the specificality of gender roles in these activities in rural areas [72]. Although the peak for fire use is the dry season for selected activities and the use of fire by women may negligible. For instance, fire use for land preparation in this study comes as the biggest fire activity, meanwhile in the Northern savanna zones of Ghana, land preparation for cropping is a preserve for men. Women do the planting and harvesting of crops as practiced in most parts of Africa [73]. Thus, gender makes a statistically significant difference in the season and reason for fire use as observed in the analyses. In this vein [74], argue that the way rural men conduct their lives has an enormous impact on how rural women live their lives. Through the division of labour, women in the north of Ghana are not responsible for land preparation for cropping hence may not be directly involved burning for cropping as but may be involved in other socio-cultural activities that use fire as some anecdotal evidence shows that women use fire to reduce grass when going for sheanut picking in the woodlands to and hance easy picking and reduce bites from snakes and other reptiles [75], using examples from Indonesia, assert that women's involvement in environmental management decisions as a whole has changed for good, over time and must be applauded. So that gender really matters when it comes to fire use and management in rural areas [72]. Since the effects of fires on the parklands of West Africa affect livelihoods of both men and women, the inclusion of men and women in the conservation of savannas is necessary.

The season and reasons for fire use and control of fires are mostly for livelihood purposes which are linked to occupation as observed in this study. The results support the argument that humans have influenced fire regimes around the world, including the African continent and West Africa specifically [68,76]. It is important to mention that people's practices, knowledge and perceptions in turn influence the reason and season of burn and the frequency of fires in particular. In the Guinea savanna of West Africa for instance, occupation drives the use of fire, which over time, impacts on vegetation, soils and other aspects of the environment at the local level, and the sub-Saharan regional level [13,41].

## Frequency of fire use for selected activities across the districts

The frequency of fire use is an important attribute of fire regimes which is characterised by the fire return interval and the number of fires that occur within a given period of time. Most of

the respondents were involved in farming as their occupation, hence, the high response rate for using fire once a year for land preparation. This supports [30] (2020) findings of frequent and high fire use on croplands in African savannas. The response for fire use once a year corresponds with the unimodal rainy season and the long dry season (up to six months) in the study area. Studies have shown that crop farmers in the West African savannas, burn to reduce trash and get rid of the grass and other herbaceous vegetation from the previous growing season. The burning is done for easy hoeing or ploughing of the land for the next cropping [77]. The respondents who indicated that they used fire twice a year for land preparation, may do this on occasions where they have to clear a new place for cropping; the first burning is to clear the bush, then fire is used again to remove stumps of the large woody species as observed by [78] and during the field reconnaissance survey for this study.

The high fire frequency districts are located in a closed savanna [79] that is characterised dominated by tall grasses and relatively good soil for agricultural production. As a result, even people in the low fire areas, particularly those that are 'urbanised', move to these areas for farming using traditional agricultural practices, which could also contribute to high fire frequency. High grass growth during the rainy season enhances the fuel load in the dry season. This results in high intensity fires with burning temperatures between 791 and 896°C as observed by [29] in the study area [80] also observed that fire return intervals are largest for tall savanna woodlands and dry forests. Thus, there is a possibility that fires in these areas may increase as conditions (heavy rainfall that promotes good grass growth and the hash Harmattan conditions) for fire ignition become more conducive [81]. Population growth in urban areas and the resultant high demand for pulses, cereals and yam grown in the agroforestry parklands (where smallholder farmers still use traditional methods of farming involving the use of fire) could also increase the use of fire for land preparation in West African savannas.

According to [82], the use of fire for weed control is another cultural practice amongst rural dwellers in most African savannas. Some respondents indicated that they used fire for weed control, including the control of *Striga hermonthica*, a parasitic plant that reduces crop yields. This was also observed in Kenya, where farmers indicated burning and hand weeding to control *Striga hermonthica* on crop fields [83–86], have also reported that the use of prescribed fires (also referred to as flame weeding) to control recalcitrant pests, weeds and invasive species such as *Centauries solstitialis* (Yellow star-thistle) in California [84]. Prescribed fire use in Ngorongoro, Kenya, was also successful in the control of ticks [87]. However, the use of weedicides and herbicides is probably gaining ground because of their ready availability in local markets even in remote areas of Ghana [88]. Some respondents explained that it is difficult to apply fire while crops are still in the field and would prefer to use chemicals during the cropping season and rather burn for land preparation. This could also be the reason for the relatively high responses for fire use after harvesting for burning stubble, which ispart of the process of land preparation and crop production [89].

Studies [6,19] have shown that the use of fire for clearing bush around homesteads during the dry season is a common practice in rural communities in Africa. Most of the study districts are rural with homes having bushy surroundings which dry out during the every Harmattan season. As a result, applying fire is much cheaper and more practical than weeding or applying weedicides, as explained by some of the respondents. In contrast, the districts in urban areas with high human populations have more paved surroundings, thus little use of fire for this activity has been observed [82]. also have shown that the use of fire in crop fields is to save on labour and the cost of chemicals [90].

Homesteads are mostly used for compound gardens in the rainy season to supply vegetables to supplement the nutritional needs of the household. These supplementary farms are known as compound farms which are burnt to get rid of the dried stubble in time for the next rainy

season. Burning around homesteads is also done to get rid of ticks and reptiles as these homesteads contain stables for cattle and other livestock [47,91]. The practice of using fire for creating firebreaks was relatively high across the study areas. This confirms the findings of [6] and [19], who observed in Mozambique and Kenya, respectively, that using fire for firebreaks was done to reduce fire risks within the communities.

Questions on hunting and charcoal burning appeared to be quite sensitive particularly in communities with government protected forests but had vast areas burnt, either due to uncontrolled fires from hunting or poaching, as well as charcoal production [11,26]. Because some people felt reluctant to indicate their use of fire for these activities, we made them to understand that the questionnaire was for study purpose. Aside from responses for questions regarding charcoal burning and hunting across the districts, anecdotal evidence, satellite observations (SERVIR West Africa, 2019) and field observations during data collection revealed that there is a high use of fire for charcoal production and hunting. We also observed a lot of on-going charcoal production in some of the communities. The respondents' reluctance in giving honest answers on their use of fire for charcoal production and hunting could be attributed to people's awareness of the dangers and adverse effects that these activities have on the environment, human lives and property. Also, there has been a radio campaign against these practices, as well as local announcements made by traditional authorities at the beginning of every dry season [92].

Additionally, the respondents seem to be aware of the suggested policy ban on charcoal production and so some of the respondents would not indicate that they used fire for charcoal burning. It is not surprising that people's perception and the frequency of fire use for this activity showed a negative correlation. We realised that although people were aware of the effects of fire on the environment, they still produce charcoal, which has become an alternative source of livelihood income in the dry season in recent times due to the unpredictable rainfall and crop failure [11], suggested that instead of placing a ban on charcoal, which is a lucrative source of livelihood income for rural people in the savannas of Ghana, the government should put in place measures to sustain the charcoal industry. However [93], argued that efforts should be made to reduce and prohibit the use of charcoal in cities and towns in order to reduce the degradation and its attendant effects on the savanna and forests in Africa.

### Fire control practice

The practice of fire control for the selected activities refers to the use of fire with caution and supervision to achieve the desired results. [6] indicated that fires that are deliberately ignited for livelihood activities are usually controlled [94] also found that traditional households that use fire in the Brazilian Amazon were actively engaged in fire management to protect their properties and farmlands.

It was observed that the high fire frequency districts had the highest responses for the use of fire once a year for almost all the selected activities. The respondents also exhibited more knowledge and practice of fire control for all the activities than those in moderate and low fire frequency districts who never tried to controlled fire. However, this may not necessarily mean that those who never control fires are not aware of the control measures, but may use less fire hence; there is little need for fire control.

The high proportion of respondents who control fire for the selected activities in the high fire frequency districts is an indication that the people were aware of the need to control fires to prevent destruction to humans, wildlife and property. This was observed in studies in Brazil [94] and Kenya [82] (Nyongesa and Vacik 2018) where fires applied intentionally, are always controlled. In Burkina Faso, fire control practice is embedded in fire management

plans and practices, which has proven successful in fire use for agriculture and customary burning [33].

## Perceptions of the importance of fire regimes and the knowledge of fire effects on the environment

It was noted that fire regimes in the Guinea savanna are characterised by land use patterns and practices linked to livelihood and socio-cultural activities, and which are in turn determined by the season of burning, severity, frequency and size. The season of burning was rated very important, which confirms why most burning is done in the dry season to prepare farm lands for cropping in the rainy season [80]. This finding agrees with other studies which found vegetation burning in other tropical savannas [39] to occur mostly in the dry season; the active fire season in the Sudano Guinean savanna. The same studies [28,13] (Rose Innes, 1972; F. K. Dwomoh and Wimberly, 2017; Laris et al., 2017) reported that most of the fires are caused by humans for the activities mentioned in this study and other studies [13,68]. Our observation during a reconnaissance survey was that most burning was done during December and January (early dry season) with a reduction in the number of fire occurrences towards the mid (February) and late dry season (March-April). Respondents' explanations indicated that burning is, however, dependent on the time the rains end (drought sets in). Thus, the timing and severity of drought within the dry season are fire conditions that play a major role in the fire regimes of the Guinea savanna.

Most respondents indicated that fire could be devastating to all aspects of the environment, (including pollution of water bodies contributing to soil erosion) if not controlled; however, they attached more importance to the season of burning over the intensity of the fire. For instance [95], also placed more emphasis on the season of burning than severity. The authors attributed the season of burning as a determinant of intensity and severity. Nonetheless these attributes are characterised by the extent of drought, which influences fuel load, moisture content and local conditions (land use patterns and practices) and vary from place to place in the Guinean savanna of West Africa. [96] and [97] also observed in Southern Africa that fire severity is determined by the season of burning which is also influenced by factors including the moisture content, fuel load and fuel characteristics. A recent study by [76] on fire behaviour in the Guinea savanna of Ivory Coast, confirmed that the severity and intensity of fire is influenced by season of burning. The study also indicated that the rate of spread and intensity of fires increased with the length of the dry season.

However, most of the respondents did not see the link between the season of burning and the severity of fire. Although the respondents indicated that they burnt earlier in the dry season when the vegetation is not too dry, they rated severity of fire as not important. This suggests a knowledge gap in respondents understanding of the concept of fire regime. This confirms [24] findings that people had limited understanding on how some fire attributes interrelate and influence one another. This supports [98] recommendations that the results of scientific research on the concept of fire regimes should be merged with traditional fire knowledge for long term integrated fire management.

Fire frequency was rated as unimportant by respondents. Contrarily, there was, however, a strong positive correlation between fire effect on the environment and how often people used fire for land preparation. the majority of respondents did not perceive fire return intervals and number of ignitions as a problem [99] argued that human-caused fire frequency (number of ignitions) is unimportant, because the onset of natural fires could have more devastating affects since natural fires occur when there is much fuel load for burning. This can result in high intensity fires than frequent human-ignited fires. However [100], found that fire

frequency has an enormous effect on both plant and animal species, and therefore should not be underestimated. In the Brazilian Cerrado, for instance [101], studied how pastoralists create seasonal mosaic patterns of burnings performed to protect fire-sensitive vegetation and avoid wildfires. However, there are some knowledge gaps in the interactions between human-caused fires, land uses and fire regimes-e.g., fire type and spread (Huffman 2013) and how these affect population structure and abundance of woody species, soils and animals in unconfined areas or the commons.

Most respondents thought the type of fire which encompasses pattern and size of fire was unimportant. However [97], have shown that fire severity is influenced by the type of fire which is in turn affected by the season of fire. This suggests the need to address the knowledge gap in managing the traditional uses of fire and fire regimes in different savanna regions in Africa.

As shown in this and other studies, most fires in these savannas occur annually and at specific times, which has defined the anthropogenic fire regimes in these savannas [27,102]. There is a need for further studies to understand the complexities of human-driven fire regimes in African savannas where season and frequency of fires are the seemingly recognisable components of fire regime by traditional fire users as shown in this and other studies [19,29,68,102,103]. The other components (type, behaviour, intensity, severity) of a fire regime have received low recognition in traditional fire knowledge management [24]. Thus, [102] asserted that the season of burning or the conditions of combustion and type of vegetation burnt could be complex, hence misinterpreted as a change in fire frequency and vice versa.

## Conclusion

The study complements other studies that indicates the use of fire is the savanna parklands of west Africa is mainly for agriculture and some socio-cultural purposes which is a characteristic of traditional livelihood practices. Most of the respondents are farmers who use fire annually for land preparation for cropping thus occupation contributed strongly to the use in the dry season. This was confirmed through the predictors with each of the response variables. The high fire frequency districts used fire for almost all the selected activities, thereby increasing the fire occurrences in these districts which aligns with the analysed fire count data obtained from the CSIR Meraka Institute.

The study however, revealed that fires that are set on purpose were mostly controlled, particularly for land preparation for cropping. This indicates people's awareness of the hazards associated with uncontrolled fires. There's the need to study the ecological effects of traditional fire use through the integration of historical and current human fire use ecology. This will facilitate a transdisciplinary approach to traditional fire management.

Generally, the season and frequency of burning was rated as a very important components of fire regimes. The respondents' stated that early and late season burning was dependent on the time the drought sets in. Notwithstanding, the type of fire was the least rated amongst the four attributes of a fire regime. More awareness on the impacts of fire frequency and different fire regimes will improve fire management in fire-prone areas as most respondents were aware of the adverse effects of fire on the environment. This positive knowledge base canalso be a foundation for the education and training in good fire management practices in the rural communities of West Africa.

The former fire volunteer squads in some communities should be retrained and replicated by district assemblies to assist in the management of fires for agricultural purposes as well as other socio-cultural practices within rural areas in the north of Ghana where fire use is common. Collaborative sharing and learning of traditional ecological fire knowledge management

between regional, district directorates of Ministry of Food and Agriculture, the Forestry Commission, the Environmental Protection Authority, Environmental NGOs and researchers is important in the management of the Guinea savanna of Ghana and West Africa at large.

## Supporting information

**S1 Appendix. Questionnaire on the study of people's perceptions and knowledge on the use and the impacts of fire on savanna ecosystems in selected communities, Northern Region, Ghana.**
(PDF)

**S2 Appendix. Fire count data for Ghana.**
(PDF)

**S1 File.**
(PDF)

**S2 File.**
(XLS)

**S3 File.**
(XLS)

**S4 File.**
(XLS)

## Acknowledgments

Many thanks to the leaders and members of the study communities for their time and support. We are very grateful to the research assistants from the Department of Agricultural Economics and Extension, University for Development Studies, Tamale, Ghana, who assisted in conducting field interviews. Our sincere appreciation is extended to Dr Amos Karbo-Bah of the University of Energy and Natural Resources, Sunyani, Ghana, who provided the first batch of data on fire counts, as well as to the CSIR Meraka Institute, South Africa, where we received a five-year data set on daily fire counts for Ghana.

## Author Contributions

**Conceptualization:** Esther Ekua Amoako.

**Methodology:** Esther Ekua Amoako.

**Project administration:** Esther Ekua Amoako.

**Supervision:** James Gambiza.

**Writing – original draft:** Esther Ekua Amoako.

**Writing – review & editing:** Esther Ekua Amoako, James Gambiza.

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
