## [Decision Letter · Decision Letter 0]

15 Dec 2020

PONE-D-20-25485

Traditional practices, knowledge and perceptions of fire use in a West African savanna parkland

PLOS ONE

Dear Dr. Amoako,

Thank you for submitting your manuscript to PLOS ONE. After careful consideration, we feel that it has merit but does not fully meet PLOS ONE’s publication criteria as it currently stands. Therefore, we invite you to submit a revised version of the manuscript that addresses the points raised during the review process.

We look forward to receiving your revised manuscript.

Kind regards,

Patrice Savadogo, PhD

Academic Editor

PLOS ONE

Journal Requirements:

2. We note that Figure 1 in your submission contains map/satellite images which may be copyrighted.

We require you to either (a) present written permission from the copyright holder to publish this figure specifically under the CC BY 4.0 license, or (b) remove the figure from your submission:

b. If you are unable to obtain permission from the original copyright holder to publish this figure under the CC BY 4.0 license or if the copyright holder’s requirements are incompatible with the CC BY 4.0 license, please either i) remove the figure or ii) supply a replacement figure that complies with the CC BY 4.0 license. Please check copyright information on all replacement figures and update the figure caption with source information. If applicable, please specify in the figure caption text when a figure is similar but not identical to the original image and is therefore for illustrative purposes only.

Additional Editor Comments:

Overall, I think that this paper makes a valuable contribution to understanding traditional fire management practices, knowledge and perceptions in a West African savanna parkland. It is based on quantitative survey. I have some reservation on the organizational and statistical questions and analysis. A mixed-method approach [research which refers to the collection and analysis of both qualitative and quantitative data in a single study in which the data are collected concurrently or sequentially, are given priority, and involve integration of the data at one or more stages in the processes of research] could have been explored in addition to using other type of statistical analyse responding to the dichotomus nature of some of the response. If theses issues are solved the authors will need to rework on the entire sections of the manuscript.

See further in:

Creswell, J. W., V. L. Plano Clark, M. L. Gutmann, and W. E. Hanson. 2003. Advanced mixed methods research designs. In A.Tashakkori & C.Teddlie (Eds.), Handbook of mixed methods in social and behavioral research (pp. 209-240). Thousand Oaks, CA: Sage.

Further, the authors will need to conduct a thorough review on the existing litterature in the savanna region of West Africa on fire management in general. There are some important piece of research which could give further insight, yet no reference is made on that.

The reviewers have made valuable comments and I invite the authors to also take that into consideration during the revision.

Reviewers' comments:

Reviewer's Responses to Questions

**Comments to the Author**

1. Is the manuscript technically sound, and do the data support the conclusions?

Reviewer #1: Partly

Reviewer #2: Yes

Reviewer #3: Partly

2. Has the statistical analysis been performed appropriately and rigorously? 

Reviewer #1: Yes

Reviewer #2: No

Reviewer #3: I Don't Know

3. Have the authors made all data underlying the findings in their manuscript fully available?

Reviewer #1: No

Reviewer #2: No

Reviewer #3: Yes

4. Is the manuscript presented in an intelligible fashion and written in standard English?

Reviewer #1: Yes

Reviewer #2: Yes

Reviewer #3: No

5. Review Comments to the Author

Reviewer #1: This paper called « Traditional practices, knowledge and perceptions of fire use in a West African savanna parkland » try to develop a work based on a quantitative survey about fires uses in Ghana. This work is well presented but it does not present original results on this topic. This paper appears as a monograph and not as a paper structured by a scientifical question with a key issue.

In details, references do not appears well used with confusion between works based in Africa, in tropics and sometimes in parklands. A lot of West African studies about fire uses are not given. Is there original characteristics for « West African savanna parkland » ? About the survey the random process used without clear justification do not allow a high quality to this survey about fire uses which is socially structured in those areas. In conclusion, a sentence is written : « There’s the need to study the ecological effects of traditional fire use » : yes we do agree but a state of the art is needed and this survey as presented here do not allow to give answers on this.

Reviewer #2: General Comments:

The ms tackles an interesting issue and a globally important one at the same time. Understanding traditional practices, knowledge and perceptions of fire use in Africa is very important for management purposes.

The reason, frequency and period of burning could help to mitigate and/or to manage bushfire in West African savanna.

The manuscript will benefit from a revision of the abstract, data analyses, and discussion sections. Most of the results show a district effect which is not discussed and not addressed in the abstract. The data analysis section is not clear, which tests are used to answer which questions? Research questions should be added at the end of the introduction for more clarity.

The number of district selected for the survey is representative (I guess) of the part that burns in Ghana and sufficient for statistical analysis, but the reasons for choosing these districts and communities are not clear.

The conclusion is too long, you have to keep the main results only.

Specific Comments:

- Most of the results (see below: lines 232, 242, 248, 253 …) show a district effect which is not dnot addressed in the abstract.

line 232: The was a significant association between the frequency of fire used for weed control and the district;

line 242 : Bush clearing around homesteads also showed strong association amongst the districts

line 248: Respondents’ use of fire to create firebreaks showed a strong significant association among the districts

line 253: Similarly, there was a significant association of fire use for hunting across the study districts.

- Line 24: add of after terms

- Line 3: add comma after thus

- Line 37: you could delete words in parenthesis and delete ‘has contributed to the transformation of landscapes over centuries and’, since you already talk about from line 30 to 31. Send the remaining part (Thus, the practice of fire use in traditional agriculture has been the source of food supply for both rural and urban economies of most countries of sub-Saharan Africa (Halstead 1987; Bassett, 1988; Holden 1993; Bassett et al. 2003)) at line 43 after (Rose Innes 1972; Archibald and Bond 2003; Dwomoh and Wimberly 2017).

- I'm not sure if it's a question of windows version but there are a lot of extra spaces in your manuscript: lines 70, 71, 72, 81, 101, 123, 124,135, 181, 188, 197, 202, 224, 225, 233, 267, …339,343, , 353, 355, …370, 371, 381, 386, 407, 427, 451, 460, 483, 486 ……..

- Line 75: Could you please add research questions here?

- From line 81 to 84: sentence too long, make the short sentences in your manuscript

- Line 88: what is the cropping season period? This period is important and have to be discuss in discussion part, since the majority of respondents indicated that they used fire for land preparation.

- Line 92: your study sites are in the northern part of Ghana? You have to precise it please.

- Line 92: the region experiences…Which region are you talking about here please?

- Line 100: how many district constitute the Northern region? Are all the districts in this northern part crossed by fire? This allows appreciating the representativeness of your study sites.

- Line 143 and figure 1: from 18 districts, why did you choose these six districts?

- Data analysis: because you did not precise research questions, it is difficult to understand which test you used to reply to which question. This part is not clear.

- There is a mistake somewhere. At line 197 you wrote “295 were from female and 237 were from male respondents women”, while in table 1 these numbers are reversed.

- Line 205: Delete 0 after 3

- Line 226: Did you applied chi-square test to reveal this association? You have to clarify in data analysis.

- Line 233: add space after Table 2.

- Line 246: the same activity, which activity please?

- Line 249 and 251: you wrote fire-break while elsewhere firebreak, used firebreak throughout the manuscript

- Line 272: I think “they controlled fire for land preparation” is not the best sentence, I suggest “they controlled fire used for land preparation” or the kind of sentence you used at line 288-290.

- Line 277: “never controlled fire used for weed control” rather than “never controlled fire for weed control”

- Line 280: add fire before districts

- Line 281: replace ‘the highest number (90%) of respondents” by “the majority of respondents (90%)

- Line 281: correct, not lowes

- Line 282: “districts never controlled fire used for burning stubble” rather than “districts never controlled fire for burning stubble”

- Line 297: Which statistical test shows this significant relationship?

- From 324 to 325: correlation test is performed between two quantitative variables, what is your variables for this correlation test here? Could you please show the table you used?

- Line 338: 4.1 rather than 4.2

- From 343 to 346: sentence too long

- Table 2: adjust the numbers in column 6 to the other rows of different columns.

- Line 365: you have to precise the main used of fire revealed by your study. Thus, when you will develop and explain the importance of the different used of fire it will be clearer.

- Line 368: replace hand –pulling by hand weeded

- Line 384: you could give the name of this district? Could you also explain why that used of fire is important in this district than in others districts? You have to discuss the difference between districts you mentioned in result section.

- Line 401: add comma after hence

- Line 402: Is it in all the districts that the population is more farmers than hunters?

- Line 403: add “.” At the end of the sentence

- From 397 à 412: Merge these 2 paragraphs in only one paragraph

- Line 404: In table 2, 55 and 43 respondents respectively in high and low fire frequency district used fire for hunting (important numbers compared to numbers relative to “never”), but you mentioned at line 401 “people would not indicate that they were involved in hunting”

And at line 404 “the low response rate for questions relative to hunting and charcoal”. I do not understand.

- Line 424 and 440: standardize the font of the titles…as at line 338 you did not used the same font.

- Line 444: add references after size.

- 485 add space after fire

- Line 509: no references in conclusion generally.

Reviewer #3: The statistical analysis was not as clear in the statements in the Discussion. Comparison to statements and Table values for use of burning for hunting was an example. The word usage was an interesting choice, perhaps differences in framing between Ghana spoken English vs. US English. There are minor but needed corrections to Font/text size, and punctuation. The figure/maps of study area need to be of better resolution/quality. Address more clearly Gender, men vs. women responses and how types of burning more more traditionally gender tasks/duties of domestic/village life and culture. Be consistent with et al. or listing all authors per Publication guidelines of journal. See yellow high lighted areas that need authors' attention/edits.

6. PLOS authors have the option to publish the peer review history of their article (what does this mean?). If published, this will include your full peer review and any attached files.

Reviewer #1: No

Reviewer #2: No

Reviewer #3: **Yes: **Frank K. Lake

---

## [Author Response · Author response to Decision Letter 0]

30 Jun 2021

Dear Editor,

Response to Academic Editor and Reviewers comments

To the best of our knowledge, we have revised the manuscript according to the comments and contribution by the reviewers and the academic Editor. The responses to the second reviewer are in review comments. The rest of the rebuttals are highlighted in blue.

All of the following will be uploaded as requested.

Journal Requirements:

 Yes, we have gone through the formatting sample and revised the according to the journal requirements.

2. We note that Figure 1 in your submission contains map/satellite images which may be copyrighted.

We require you to either (a) present written permission from the copyright holder to publish this figure specifically under the CC BY 4.0 license, or (b) remove the figure from your submission: 

 I have attached communication between SERVIR/NASA West Africa which explains that we acknowledge the source. I have also done this by indicating the sources of the data. If this is not allowed, I will send the forms to SERVIR. 

The map on fire gradient was created by the authors using fire count data obtained from CSIR Meraka Institute and EORIC, Ghana. All has been duly acknowledged. I would however want to know if I have to ask permission from EORIC and CSIR Meraka Institute for using the fire count data to create the fire map. 

Additional Editor Comments:

Overall, I think that this paper makes a valuable contribution to understanding traditional fire management practices, knowledge and perceptions in a West African savanna parkland. It is based on quantitative survey. I have some reservation on the organizational and statistical questions and analysis. A mixed-method approach [research which refers to the collection and analysis of both qualitative and quantitative data in a single study in which the data are collected concurrently or sequentially, are given priority, and involve integration of the data at one or more stages in the processes of research] could have been explored in addition to using other type of statistical analyse responding to the dichotomus nature of some of the response. If theses issues are solved the authors will need to rework on the entire sections of the manuscript.

See further in:

Creswell, J. W., V. L. Plano Clark, M. L. Gutmann, and W. E. Hanson. 2003. Advanced mixed methods research designs. In A.Tashakkori & C.Teddlie (Eds.), Handbook of mixed methods in social and behavioral research (pp. 209-240). Thousand Oaks, CA: Sage.

Further, the authors will need to conduct a thorough review on the existing litterature in the savanna region of West Africa on fire management in general. There are some important piece of research which could give further insight, yet no reference is made on that.

We found a few grey literature on fire management in West Africa as well as ahandbook onfire management in Sub-Saharan Africa but focus on Southern African contexts Most of the published peer reviewed papers have already been used eg Bassett (2003), Boffa (1995), Kugbe et al (2014). However we have included a bit of review and characteristics of the West African savanna in the introduction

The reviewers have made valuable comments and I invite the authors to also take that into consideration during the revision.

Reviewers' comments:

5. Review Comments to the Author

Reviewer #1: This paper called « Traditional practices, knowledge and perceptions of fire use in a West African savanna parkland » try to develop a work based on a quantitative survey about fires uses in Ghana. This work is well presented but it does not present original results on this topic. 

Please what is the original results referred to here - is not well understood. We did the field survey and this a bit of the results from the survey. We cannot publish all the results in one paper. So, we decided to publish this on perception. We plan to publish fire use and land use and the last bit on fire management.

This paper appears as a monograph and not as a paper structured by a scientifical question with a key issue. 

We have included the questions in the introduction.

In details, references do not appears well used with confusion between works based in Africa, in tropics and sometimes in parklands. A lot of West African studies about fire uses are not given. 

We have references on Africa because we want to look at fire in the broader context of savanna and the tropics because most part of Africa and for that matter West Africa is in the tropics. We have however modified to emphasize on West Africa and Parklands.

Is there original characteristics for « West African savanna parkland »? 

The major characteristics are highlighted in Lns 45 -52.

About the survey the random process used without clear justification do not allow a high quality to this survey about fire uses which is socially structured in those areas. In conclusion, a sentence is written: « There’s the need to study the ecological effects of traditional fire use » : yes we do agree but a state of the art is needed and this survey as presented here do not allow to give answers on this.

The reason for the selection of the districts and communities has been explained in Ln 134 -161.

Reviewer #2: General Comments:

The ms tackles an interesting issue and a globally important one at the same time. Understanding traditional practices, knowledge and perceptions of fire use in Africa is very important for management purposes.

The reason, frequency and period of burning could help to mitigate and/or to manage bushfire in West African savanna.

The manuscript will benefit from a revision of the abstract, data analyses, and discussion sections. Most of the results show a district effect which is not discussed and not addressed in the abstract. The data analysis section is not clear, which tests are used to answer which questions? Research questions should be added at the end of the introduction for more clarity.

We have added the research questions.

The number of district selected for the survey is representative (I guess) of the part that burns in Ghana and sufficient for statistical analysis, but the reasons for choosing these districts and communities are not clear.

Yes that we wanted to be near if not representative (6 out of 18 but now 26 districts) The reason for selecting the districts have been explained in the methods section. 

The conclusion is too long, you have to keep the main results only.

The conclusion has been trimmed.

Specific Comments:

- Most of the results (see below: lines 232, 242, 248, 253 …) show a district effect which is not dnot addressed in the abstract.

This has been addressed in the abstract - A sentence has been added to give some explanation .of the selected districts

line 232: The was a significant association between the frequency of fire used for weed control and the district;

line 242 : Bush clearing around homesteads also showed strong association amongst the districts

line 248: Respondents’ use of fire to create firebreaks showed a strong significant association among the districts

line 253: Similarly, there was a significant association of fire use for hunting across the study districts.

We could not include all the results in the abstract because of the word limit of abstracts in general. Thus, we highlighted the results for the very common use of fire as revealed in the study.

- Line 24: add of after terms

- Line 3: add comma after thus

- Line 37: you could delete words in parenthesis and delete ‘has contributed to the transformation of landscapes over centuries and’, since you already talk about from line 30 to 31. Send the remaining part (Thus, the practice of fire use in traditional agriculture has been the source of food supply for both rural and urban economies of most countries of sub-Saharan Africa (Halstead 1987; Bassett, 1988; Holden 1993; Bassett et al. 2003)) at line 43 after (Rose Innes 1972; Archibald and Bond 2003; Dwomoh and Wimberly 2017).

We have reworked the sentence to conform to the reviewers’ suggestion.

- I'm not sure if it's a question of windows version but there are a lot of extra spaces in your manuscript: lines 70, 71, 72, 81, 101, 123, 124,135, 181, 188, 197, 202, 224, 225, 233, 267, …339,343, , 353, 355, …370, 371, 381, 386, 407, 427, 451, 460, 483, 486 ……..

This has been resolved.

- Line 75: Could you please add research questions here?

Research questions have been added to the Introduction.

- From line 81 to 84: sentence too long, make the short sentences in your manuscript

Yes, this has been resolved.

- Line 88: what is the cropping season period? This period is important and have to be discuss in discussion part, since the majority of respondents indicated that they used fire for land preparation. 

The cropping period has been indicated.

- Line 92: your study sites are in the northern part of Ghana? You have to precise it please.

The study site has been indicated.

- Line 92: the region experiences…Which region are you talking about here please?

Please this has been indicated.

- Line 100: how many district constitute the Northern region? Are all the districts in this northern part crossed by fire? This allows appreciating the representativeness of your study sites. 

There were originally 18 districts which but been increased to 26. However, the district are not new districts but old one that have been partitioned. Thus, fire use in the districts will be the same.

- Line 143 and figure 1: from 18 districts, why did you choose these six districts?

Explained in methods section now.

- Data analysis: because you did not precise research questions, it is difficult to understand which test you used to reply to which question. This part is not clear.

The research questions have been incorporated.

- There is a mistake somewhere. At line 197 you wrote “295 were from female and 237 were from male respondents women”, while in table 1 these numbers are reversed.

This was an oversight. It has been resolved.

- Line 205: Delete 0 after 3

- Line 226: Did you applied chi-square test to reveal this association? You have to clarify in data analysis. 

Yes, we indicated in data analysis section.

- Line 233: add space after Table 2. fixed

- Line 246: the same activity, which activity please? Indicated now

- Line 249 and 251: you wrote fire-break while elsewhere firebreak, used firebreak throughout the manuscript Changes have been made

- Line 272: I think “they controlled fire for land preparation” is not the best sentence, I suggest “they controlled fire used for land preparation” or the kind of sentence you used at line 288-290.

- Line 277: “never controlled fire used for weed control” rather than “never controlled fire for weed control”

- Line 280: add fire before districts

- Line 281: replace ‘the highest number (90%) of respondents” by “the majority of respondents (90%)

- Line 281: correct, not lowes

 - Line 282: “districts never controlled fire used for burning stubble” rather than “districts never controlled fire for burning stubble” 

- Line 297: Which statistical test shows this significant relationship?

- From 324 to 325: correlation test is performed between two quantitative variables, what is your variables for this correlation test here? Could you please show the table you used?

This information is part of a bulky data for the whole survey so I selected the variables to run the analysis in SPSS. The Spreadsheet with this information is quite bulky to share.

- Line 338: 4.1 rather than 4.2

- From 343 to 346: sentence too long

- Table 2: adjust the numbers in column 6 to the other rows of different columns.

- Line 365: you have to precise the main used of fire revealed by your study. Thus, when you will develop and explain the importance of the different used of fire it will be clearer.

- Line 368: replace hand –pulling by hand weeded

- Line 384: you could give the name of this district? Could you also explain why that used of fire is important in this district than in others districts? You have to discuss the difference between districts you mentioned in result section. 

- Line 401: add comma after hence

- Line 402: Is it in all the districts that the population is more farmers than hunters? 

- Line 403: add “.” At the end of the sentence

 - From 397 à 412: Merge these 2 paragraphs in only one paragraph

- Line 404: In table 2, 55 and 43 respondents respectively in high and low fire frequency district used fire for hunting (important numbers compared to numbers relative to “never”), but you mentioned at line 401 “people would not indicate that they were involved in hunting”

And at line 404 “the low response rate for questions relative to hunting and charcoal”. I do not understand.

- Line 424 and 440: standardize the font of the titles…as at line 338 you did not used the same font.

 - Line 444: add references after size.

- 485 add space after fire

- Line 509: no references in conclusion generally. 

Reviewer #3: The statistical analysis was not as clear in the statements in the Discussion. Comparison to statements and Table values for use of burning for hunting was an example. The word usage was an interesting choice, perhaps differences in framing between Ghana spoken English vs. US English. There are minor but needed corrections to Font/text size, and punctuation. The figure/maps of study area need to be of better resolution/quality. Address more clearly Gender, men vs. women responses and how types of burning more more traditionally gender tasks/duties of domestic/village life and culture. Be consistent with et al. or listing all authors per Publication guidelines of journal. See yellow high lighted areas that need authors' attention/edits. 

It is our hope that we have been able to respond appropriately to the points/comments raised by the reviewers and the Academic Editor.

Kind regards,

Esther

---

## [Decision Letter · Decision Letter 1]

21 Feb 2022

PONE-D-20-25485R1Traditional practices, knowledge and perceptions of fire use in a West African savanna parklandPLOS ONE

Dear Dr. Amoako,

Thank you for submitting your manuscript to PLOS ONE. After careful consideration, we feel that it has merit but does not fully meet PLOS ONE’s publication criteria as it currently stands. Therefore, we invite you to submit a revised version of the manuscript that addresses the points raised during the review process. Revised as per suggestions.

We look forward to receiving your revised manuscript.

Kind regards,

Randeep Singh

Academic Editor

PLOS ONE

Reviewers' comments:

Reviewer's Responses to Questions

**Comments to the Author**

1. If the authors have adequately addressed your comments raised in a previous round of review and you feel that this manuscript is now acceptable for publication, you may indicate that here to bypass the “Comments to the Author” section, enter your conflict of interest statement in the “Confidential to Editor” section, and submit your "Accept" recommendation.

Reviewer #4: (No Response)

Reviewer #5: (No Response)

2. Is the manuscript technically sound, and do the data support the conclusions?

Reviewer #4: No

Reviewer #5: Partly

3. Has the statistical analysis been performed appropriately and rigorously? 

Reviewer #4: No

Reviewer #5: Yes

4. Have the authors made all data underlying the findings in their manuscript fully available?

Reviewer #4: No

Reviewer #5: (No Response)

5. Is the manuscript presented in an intelligible fashion and written in standard English?

Reviewer #4: Yes

Reviewer #5: Yes

6. Review Comments to the Author

Reviewer #4: Plos One_D-20-25485R1

Traditional practices, knowledge and perceptions of fire use in a West African savanna parkland—Esther Amoako, James Gambiza

A potentially interesting and useful study addressing cultural fire management practices in culturally diverse agricultural and agroforestry settings of northern Ghana. However, I don’t think the current format and focus of the paper adequately nor usefully explores the assembled data and, from my reading of the ms, I think a total re-analysis and rewrite is required. I apologise in advance for this assessment but trust that the comments below may assist in developing a very useful, internationally relevant study.

First up, I don’t understand why you stratified your study sites in the Northern Region by fire hotspot frequency (as an index of fire activity) when a more logical approach would be to stratify firstly by major cultural grouping (e.g. Mangrusi, Dagomba, Gonja…), noting that 92% of your responses came from these three groups, in order to explore similarities / differences in cultural fire management practices. Are your hotspot frequency classes (high, medium, low) reflective of the cultural fire management activities of such cultural groupings, and/or population density generally, or some other factor(s)? Given that the title of your ms purports to describe traditional practices, knowledge(s) and perceptions…, reframing the focus of your analysis would seem to me to make much more sense. Such an approach would also presumably help tease out some intriguing findings in your results; for example, I found it interesting that very little ‘fire control’ is practised in association with hunting, nor the implementation of firebreaks (Table 3)—doesn’t this lead to major wildfire problems and associated significant socio-cultural-economic issues? Does this reflect traditional practice in all cultural groupings?

Taking such an approach would provide more useful cultural contrasts than your current use of data stratified by hotspot fire frequency as presented in Tables 2,3,4—and I confess I don’t know what Table 5 was attempting to convey. Note also:

• any analysis of hotspot frequency data should be presented in Results, not in Methods as currently

• you need to provide a copy of the questionnaire either as an appendix or in supplementary materials, including only those questions that you use for analysis of results

• Figures need to be revisited and better presented, noting that Fig 1 is included again as Fig 1A. Given comments above why not include a map of the 5 Chiefdoms, especially to highlight the three for which you have adequate survey data

Reviewer #5: I give some specific suggestions below, but overall I am concerned that the paper is somewhat rambling and lacking a focus. It appears that the premise of the paper is that there is some objective impact of the fire regimes on the environment and interviews were conducted to see how close the respondants perceptions matched that objective reality. This is implicit in the conclusion about the benefit of education and training (line 564). However the objective reality is not spelt out clearly and the line up against interview findings is laid out in a rambling hard-to-follow way. I would really like to see this relationship tightened up considerably. It is also possible, if not probable that the objective reality is actually a perception by a different group of people and may not be as real as implied. This needs to be addressed.

Line 84: The statement of objectives really doesn’t encapsulate the description of the survey questions described in line 178-192. Please rewrite.

Line 105: “a population of nearly 2.5 million people in xxx km2, representing…”

Line 133: Were the hired Fulani interviewed? What proportion of the population are they? What proportion of the work do they do? Do they perform the work of burning? Could it be that the landowners’ perceptions about fire are different from those of the people who actually do the burning if they are hired workers?

Line 169: “People were asked questions in their respective first languages on…”

Line 207: please replace “dialects” with “languages”. The word “dialects” has racist overtones as it is far more frequently used for people of colour than of white people for their native language. (to be clear in popular usage white people speak languages, black people speak dialects - this is a racist word usage even if people don't realise it)

Line 213: Full stop required after statistics.

Line 230: 21% is about one fifth. That is not a majority. “of six age groups, the largest was 26-32 years old”

Line 238: 70 persons per household to my understanding is a small village. Please explain your definition of a household. I doubt all 70 sleep under the same roof. Do you use this information on household size? If not maybe delete sentence as irrelevant.

Line 403: Although you are citing a paper with these temperatures, they are really meaningless in terms of describing fire behaviour. There spatial and temporal variations in fire behaviour between early and late season are such that ascribing temperature values is invalid. Maybe just say that late fires are of higher intensity than early fires.

Line 414: be good to point out that Striga is a parasitic plant that reduces crop yields.

Line 442 “respectively, that firebreaks were burnt to reduce fire risks…” This expression reduces wordiness.

Line 490: “occur”

Line 540: the statement “could have enormous impacts on the environment” is rather meaningless. The intention of controlled burning is to impact the environment through such processes as removing stubble, weeds, fire hazard around dwellings, and causing game to move. What are these “enormous impacts” and do they arise from a single event (e.g. death of rainforest trees, or decline in air quality) or as a result of the fire regime (e.g. soil degradation, shift in tree/grass balance). If they are the result of a regime, then maybe they are not enormous. Are there positive environmental impacts of fires?

7. PLOS authors have the option to publish the peer review history of their article (what does this mean?). If published, this will include your full peer review and any attached files.

Reviewer #4: No

Reviewer #5: No

---

## [Author Response · Author response to Decision Letter 1]

7 Apr 2022

Dear Editor, 

Response to Reviewers’ Comments

 To the best our knowledge, we have revised the manuscript in response to the comments and contribution by the Reviewers and the Academic Editor. The responses to the reviewers and rebuttals are highlighted in blue. I have uploaded a Zip File of all the fire count data used for analysis and added the questionnaire at the end of the manuscript.

The following will be uploaded as requested: 

Responses and rebuttals to reviewers’ comments are presented below:

First up, I don’t understand why you stratified your study sites in the Northern Region by fire hotspot frequency (as an index of fire activity) when a more logical approach would be to stratify firstly by major cultural grouping (e.g., Manprusi, Dagomba, Gonja…), noting that 92% of your responses came from these three groups, in order to explore similarities / differences in cultural fire management practices.

The study did not focus on cultural groupings. The various ethnic groupings in the north of Ghana do not have different fire management strategies. If we were comparing the savanna to the transition ecological zone or the coastal savanna of Ghana then we could do the cultural comparison. The Guinea savanna was chosen because it has recorded more fire occurrences than the south of Ghana. Thus, the import of the study was to ascertain people perception of fire use and fire practices, how the frequent fires affect economic species and soils in the area. Yes, we agree to your suggestion but again cultural groupings or ethnicity, did not make significant prediction to the reason for the use of fire, indicating that all the ethnic groups have similar reasons for the use of fire, especially for land preparation for cropping which is the major reasons for fire use across all the districts. Also, we did find the Dagomba ethnic in almost every district, although they may not be indigenes of the districts. It is difficult to directly link fire use to a particular cultural grouping in this case due to the heterogeneous nature of most of the district. 

Are your hotspot frequency classes (high, medium, low) reflective of the cultural fire management activities of such cultural groupings, and/or population density generally, or some other factor(s)? 

We considered hotspot because of the high fire occurrence during the dry season in the Guinea savanna. The fire count data showed varying fire counts in the various district hence the grouping into low medium and high fire frequency Zones. It is difficult to make clear distinctions in cultural fire management because they have similar practices. The reason why we find more people use fire for almost the same activities.

Given that the title of your ms purports to describe traditional practices, knowledge(s) and perceptions…, reframing the focus of your analysis would seem to me to make much more sense. Such an approach would also presumably help tease out some intriguing findings in your results; for example, I found it interesting that very little ‘fire control’ is practised in association with hunting, nor the implementation of firebreaks (Table 3)—doesn’t this lead to major wildfire problems and associated significant socio-cultural-economic issues? Does this reflect traditional practice in all cultural groupings? 

No, we did not observe any difference in the vein. The differences in location in location or district is that some district have more fauna so they do hunt these animals in the dry season. However, people from other districts within the region move to hunt in these areas. This MS is a Chapter in a thesis which assessed the impact of frequent fires on woody species and soils in these fire hotspots in the Guinea savanna. That is why we wanted to find out what reasons and perceptions do people burn in order to give a justification of fire hotspots in the Guinea savanna.

Taking such an approach would provide more useful cultural contrasts than your current use of data stratified by hotspot fire frequency as presented in Tables 2,3,4—

 I don’t know how you will consider this, but again our understanding of and focus on traditional fire use and management is more of using fire use without much regulation like using fire for farming, hunting, burning stubble after cropping with less attribution to cultural groupings. For instance, Huffman’s (2013) review on traditional fire management showed that attributes such as type and pattern of fire are less considered in the use of fire traditional fire management. In contrast fire applied in conservation sites such as the Mole National Park are well planned.

and I confess I don’t know what Table 5 was attempting to convey. 

 Table has been deleted.

Note also:

• any analysis of hotspot frequency data should be presented in Results, not in Methods as currently

Please it has been summarized in a table and has been provided under results.

You need to provide a copy of the questionnaire either as an appendix or in supplementary materials, including only those questions that you use for analysis of results.

Please find attached

• Figures need to be revisited and better presented, noting that Fig 1 is included again as Fig 1A. Given comments above why not include a map of the 5 Chiefdoms, especially to highlight the three for which you have adequate survey data. 

We hope we have been able to explain the reason why we chose the hotspots for the analysis. Again, this paper is a Chapter in a thesis which assessed the impact of fire on woody species and soil in these fire hotspots. We can probably consider the cultural groupings in fire use in future by comparing the cultural/traditional fire knowledge in North to the south of Ghana. These have distinct cultural practices which is determined by factors such rainy season, which is unimodal in the North and bimodal in the South of Ghana.

Reviewer #5: I give some specific suggestions below, but overall, I am concerned that the paper is somewhat rambling and lacking a focus. It appears that the premise of the paper is that there is some objective impact of the fire regimes on the environment and interviews were conducted to see how close the respondents perceptions matched that objective reality. This is implicit in the conclusion about the benefit of education and training (line 564).

 However the objective reality is not spelt out clearly and the line up against interview findings is laid out in a rambling hard-to-follow way. I would really like to see this relationship tightened up considerably. 

The objective of the research has been clearly spelt out in the Introduction. Before this feedback from PLOS ONE we thought that a Multiple regression analysis would unravel some hidden details such as activities that fire is used for, reasons for fire use and fire use in the dry season and district, ethnic group, gender, age and education etc. I hope that the Multiple regression analysis adds to making a better analysis and presentation of the objectives of the research questions. Thus we changed the title to reflect data analysis and the results.

It is also possible, if not probable that the objective reality is actually a perception by a different group of people and may not be as real as implied. This needs to be addressed.

Line 84: The statement of objectives really doesn’t encapsulate the description of the survey questions described in line 178-192. Please rewrite.

This has been rewritten to encapsulate the purpose of the survey.

Line 105: “a population of nearly 2.5 million people in xxx km2, representing…”

This has been changed 

Line 133: Were the hired Fulani interviewed? What proportion of the population are they? 

Yes, the few (2%) were interviewed. Their number of Fulani in Ghana are not known, but they are estimated to be more than 14,000. They are in minority and do live in isolated areas round the outskirts of the local communities. so that it difficult to have their exact numbers. This is because they are normally nomadic and gradually becoming sedentary in some areas of the Northern region.

What proportion of the work do they do? They take care of cattle in the community thus communities members com together and hire a Fulani as herdsman. Do they perform the work of burning? 

 The study did not specifically consider that who performs burning.

Could it be that the landowners’ perceptions about fire are different from those of the people who actually do the burning if they are hired workers? 

People who burn for cropping are the same people who prepare the land for planting which normally done by men in the house hold or farm family likewise the Pastoralist. But the fact is that there no specific designated areas for grazing and farming. During the dry season the farms are also used as grazing fields in addition to the areas within the savanna as the natural pastures.

We did not consider who does the burning, we were interested in whether fire used or not, If yes, what time of the year, how many times and reasons for the fire use.

Line 169: “People were asked questions in their respective first languages on…”

Corrected

Line 207: please replace “dialects” with “languages”. The word “dialects” has racist overtones as it is far more frequently used for people of colour than of white people for their native language. (to be clear in popular usage white people speak languages, black people speak dialects - this is a racist word usage even if people don't realise it).

Well, noted and replaced.

Line 213: Full stop required after statistics. 

Done

Line 230: 21% is about one fifth. That is not a majority. “of six age groups, the largest was 26-32 years old”

This was deleted and corrected before the comments came in so may not appear in track changes.

Line 238: 70 persons per household to my understanding is a small village. Please explain your definition of a household. I doubt all 70 sleeps under the same roof.

 In the north of Ghana, a traditional household refers to a big house with several rooms belonging to the sons of (with the wives and children) one father or grandfather who is normally the head of the household. The house is normally built in circular style with a big shared compound being the centre of the house. The size of the household normally depends of number of sons who live with the father. We did not ask of the number of roofs or “apartments” in the compound but the number of people in the large compound house. The members of the house normally eat from the same kitchen and cooking pots.

Do you use this information on household size? If not maybe delete sentence as irrelevant.

We have not used but we though may be relevant, gives an idea on how many people live a household may have a direct relationship since the number and size of households are dependent determine the size and number of crops farms the family controls. Regression analysis actually did not predict for the household size.

Line 403: Although you are citing a paper with these temperatures, they are really meaningless in terms of describing fire behaviour. There spatial and temporal variations in fire behaviour between early and late season are such that ascribing temperature values is invalid. May be just say that late fires are of higher intensity than early fires. 

Yes, we deleted as suggested, but some studies in the Guinea savanna have shown that early burning have higher temperature than late burning. So, we thought it might be fine to be specific here.

Line 414: be good to point out that Striga is a parasitic plant that reduces crop yields.

Line 442 “respectively, that firebreaks were burnt to reduce fire risks…” 

This has been rephrased

Line 490: “occur”

Line 540: the statement “could have enormous impacts on the environment” is rather meaningless. The intention of controlled burning is to impact the environment through such processes as removing stubble, weeds, fire hazard around dwellings, and causing game to move. What are these “enormous impacts” and do they arise from a single event (e.g., death of rainforest trees, or decline in air quality) or as a result of the fire regime (e.g., soil degradation, shift in tree/grass balance). If they are the result of a regime, then maybe they are not enormous. Are there positive environmental impacts of fires? This has been rephrased. 

THANKS FOR THE COMMENTS

---

## [Editor Report · Decision Letter 2]

21 Apr 2022

FIRE USE PRACTICES, KNOWLEGE AND PERCEPTIONS IN A WEST AFRICAN SAVANNA PARKLAND

PONE-D-20-25485R2

Dear Dr. Esther,

We’re pleased to inform you that your manuscript has been judged scientifically suitable for publication and will be formally accepted for publication once it meets all outstanding technical requirements.

Kind regards,

Randeep Singh

Academic Editor

PLOS ONE
---

## [Editor Report · Acceptance letter]

29 Apr 2022

PONE-D-20-25485R2 

FIRE USE PRACTICES, KNOWLEDGE AND PERCEPTIONS IN A WEST AFRICAN SAVANNA PARKLAND 

Dear Dr. Amoako:

I'm pleased to inform you that your manuscript has been deemed suitable for publication in PLOS ONE. Congratulations! Your manuscript is now with our production department. 

Kind regards, 

on behalf of

Dr. Randeep Singh 

Academic Editor

PLOS ONE